# Postnatal infection surveillance by telephone in Dar es Salaam, Tanzania: An observational cohort study

**Susannah L. Woodd**[1]*, **Abdunoor M. Kabanywanyi**[2], **Andrea M. Rehman**[1], **Oona M. R. Campbell**[1], **Asila Kagambo**[2], **Warda Martiasi**[2], **Louise M. TinaDay**[1], **Alexander M. Aiken**[1], **Wendy J. Graham**[1]

**1** Department of Infectious Disease Epidemiology, London School of Hygiene and Tropical Medicine, London, United Kingdom, **2** Department of Health Systems, Impact Evaluation and Policy, Ifakara Health Institute, Dar es Salaam, Tanzania

* susannah.woodd@lshtm.ac.uk

## Abstract

### Introduction

Maternal and newborn infections are important causes of mortality but morbidity data from low- and middle-income countries is limited. We used telephone surveillance to estimate infection incidence and risk factors in women and newborns following hospital childbirth in Dar es Salaam.

### Methods

We recruited postnatal women from two tertiary hospitals and conducted telephone interviews 7 and 28 days after delivery. Maternal infection (endometritis, caesarean or perineal wound, or urinary tract infection) and newborn infection (umbilical cord or possible severe bacterial infection) were identified using hospital case-notes at the time of birth and self-reported symptoms. Adjusted Cox regression models were used to assess the association between potential risk-factors and infection.

### Results

We recruited 879 women and interviewed 791 (90%). From day 0–7, 6.7% (49/791) women and 6.2% (51/762) newborns developed infection. Using full follow-up data, the infection rate was higher in women with caesarean childbirth versus women with a vaginal delivery (aHR 1.93, 95%CI 1.11–3.36). Only 24% of women received pre-operative antibiotic prophylaxis before caesarean section. Infection was higher in newborns resuscitated at birth versus newborns who were not resuscitated (aHR 4.45, 95%CI 2.10–9.44). At interview, 66% (37/56) of women and 88% (72/82) of newborns with possible infection had sought health-facility care.

### Conclusions

Telephone surveillance identified a substantial risk of postnatal infection, including cases likely to have been missed by hospital-based data-collection alone. Risk of maternal

**Data Availability Statement:** According to Tanzanian ethics guidelines as reflected in the project data management plan and data transfer

agreement, it is not recommended to share potentially sensitive individual-level data, even after attempting to de-identify it. Therefore, anyone wishing to access this data will need to request written consent from Ifakara Health Institute who own the data. The data can be accessed from the IHI repository https://data.ihi.or.tz/index.php/catalog/edit/1. Anyone requesting the data will be required to submit a brief abstract narrating the reasons for requesting access to the data. The system will send an email to the data team and the local PI.

**Funding:** SLW, AMK, AK, WM and WJG received personal salary funded by The Soapbox Collaborative for their work on the study. Ifakara Health Institute received funding by The Soapbox Collaborative for carrying out data collection. https://www.lshtm.ac.uk/research/centres/march-centre/soapbox-collaborative SLW was supported by a Research Degree Travel Scholarship from LSTHM. AMR is additionally funded by the UK Medical Research Council (MRC) and the UK Department for International Development (DFID) under the MRC/DFID Concordat agreement which is also part of the EDCTP2 programme supported by the European Union, Grant Ref:MR/R010161/1 The funders had no role in study design, data collection and analysis, decision to publish, or preparation of the manuscript.

**Competing interests:** I have read the journal's policy and the authors of this manuscript have the following competing interests: SLW, AMK, AK, WM and WJG received personal salary funded by The Soapbox Collaborative for their work on the study. SLW received a Research Degree Travel Grant from LSTHM. AMR received salary from the UK Medical Research Council and the UK Department for International Development (DFID) under the MRC/DFID Concordat agreement. Other authors declare no competing interests. This does not alter our adherence to PLOS ONE policies on sharing data and materials.

endometritis and newborn possible severe bacterial infection were consistent with other studies. Caesarean section was the most important risk-factor for maternal infection. Improved implementation of pre-operative antibiotic prophylaxis is urgently required to mitigate this risk.

## Introduction

Preventing maternal and newborn infections is a high priority in the World Health Organization's (WHO) vision of good quality care for pregnant women and newborns [1]. Pregnancy-related sepsis is estimated to cause 11% of maternal mortality [2] and infection is responsible for 23% of newborn deaths [3] with the vast majority in low- and middle-income countries (LMICs). Increasing health-facility births in LMICs [4] presents an opportunity to reduce disease incidence through strengthened infection prevention initiatives.

Despite the importance of maternal and newborn infection, we have limited knowledge of the frequency in high-burden countries. A systematic review of maternal peripartum infection included only seven sub-Saharan Africa (SSA) studies (one from Tanzania [5]) and none were considered high quality [6]. From meta-analysis, the regional estimate for endometritis was 1.7% and for wound infection was 3.4%. A systematic review of possible severe bacterial infection (pSBI) using the Young Infant Clinical Signs Study (YICSS) criteria [7] estimated 6.2% of newborns in SSA were affected (six studies, none from Tanzania). The case-fatality risk was 14.1% [8].

The majority of severe maternal infections occur postpartum, arising from the genito-urinary tract or wounds [9, 10], and presenting after the woman has been discharged home following childbirth [11]. The majority of newborn deaths from infection occur after the first week of life [3]. Community follow-up is therefore necessary to capture all cases of infection. Home visits are resource intensive, consequently many studies only report infection up to the time of hospital discharge following facility childbirth. Mobile telephone surveillance is a possible alternative, with emerging evidence of feasibility and validity to monitor surgical site infection (SSI) in SSA [12, 13], and postnatal outcomes in India [14].

Responding to the limited data on maternal newborn infection incidence in SSA our observational cohort study aimed to estimate the incidence and risk factors for infection in women and newborns in the four weeks following hospital childbirth in urban Tanzania, using hospital case-notes from the time of birth and telephone surveillance. We also assessed the feasibility of mobile telephone assessment for infection, described care-seeking behaviour following infection and explored possible consequences of infection; hospital readmission, depression and reduced maternal function.

## Methods

This study was a collaboration between London School of Hygiene and Tropical Medicine (LSHTM) and Ifakara Health Institute (IHI) and based at two of the three public Regional Referral Hospitals in Dar es Salaam; Amana (Ilala district) and Temeke (Temeke district). Each hospital conducts approximately 1,000 births per month. It was a sub-study of a pilot evaluation of training in environmental cleaning [15].

Two research nurses per hospital recruited eligible women from postnatal wards every Monday to Thursday. They sampled from all women who gave birth in the previous 24 hours using a random number application [16] with probability proportional to delivery mode (caesarean or vaginal). Eligible women were aged 18 years or older with access to at least one

mobile telephone and providing signed or witnessed thumbprint consent. Women admitted to the intensive care unit were ineligible. Women provided up to three mobile telephone numbers; one or two of their own and one for a relative or neighbour. Replacements were sampled in the same way when potential participants were unavailable or ineligible.

Two research nurses at IHI offices in Dar es Salaam interviewed each woman twice by telephone in Kiswahili, starting seven and 28 days after recruitment. Nurses made four telephone call attempts, over seven days, to reach each woman.

## Outcomes and exposures

The primary outcomes were 1) possible maternal postnatal infection (one or more of caesarean surgical site infection, urinary tract infection, perineal wound infection or endometritis) and 2) possible newborn infection (either of pSBI or umbilical cord infection). Each outcome was measured as a rate, and as the day 7 (early infection) and day 8–28 cumulative risk. Infections were identified from women's hospital case-notes around the time of childbirth or from self-reported symptoms during telephone interview using standard definitions [7, 17, 18]. These definitions were adapted by the first author to include only symptoms and signs easily reported by the women (Table 1). Secondary outcomes were each individual infection listed above, plus mastitis.

Potential risk factors were extracted from hospital case-notes; maternal age, gestational age, parity, HIV, diabetes, hypertensive disorder, haemorrhage, prelabour rupture of membranes (PROM), induction of labour, delivery mode, postpartum haemorrhage (PPH) and infection during labour. Possible consequences of infection collected during telephone interview were self-reported readmission, depression assessed using a validated 5-question modified Edinburgh Postnatal Depression Scale (EPDS) and functionality according to five common postpartum activities (S1 Appendix).

## Data collection

Data was entered on tablets with Open Data Kit (ODK), using unique identification (ID) numbers to maintain confidentiality. Data was extracted from maternal paper case-notes after hospital discharge, including demographics, pregnancy and childbirth history, infection diagnosed during admission and antibiotics prescribed (S2 Appendix). Telephone interviews with women consisted of pre-coded closed questions on the history of specific symptoms of infection, day of symptom onset, care-seeking behaviour, and readmission to hospital. At day-28, women were also asked questions on depression and function (S1 Appendix). Women with infection symptoms were advised to attend a health-facility if they hadn't already. In cases of maternal depression or neonatal death, women were offered referral to social welfare liaison for counselling and support.

Research nurses received six days training in recruitment and data collection, including two days at the hospitals when they piloted the tools on 24 women. Telephone interview nurses additionally conducted pilot interviews with the same 24 women over two days.

## Study size

With 900 women and an estimated 10% loss to follow-up at day-28, we would have 95% confidence to estimate a maternal infection risk of 3%±1.2% with 80% power. Our daily recruitment target was 12–20 women per hospital.

## Data management

Data was cleaned and analysed using STATA 16. Gestational age was grouped as preterm (<37 weeks) or term (37–42 weeks). The depression score was grouped as no depression (0–5)

**Table 1. Syndromic infection definitions used.**

| Infection | Questions to women | Definition | Standard definition adapted |
|---|---|---|---|
| Caesarean Section Surgical Site Infection (SSI) | At the site of your caesarean section (cut/operation on your abdomen) have you experienced: | Either I. OR, (V. AND one or more of II-IV.), OR two or more of VI-VIII | CDC[a] |
| | I. Pus discharge | | |
| | II. Pain | | |
| | III. Swelling | | |
| | IV. Redness | | |
| | V. Wound breakdown (wound edges separated) | | |
| | Have you experienced: | | |
| | VI. Fever | | |
| | VII. Abdominal pain | | |
| | VIII. Foul-smelling or pus vaginal discharge | | |
| Urinary Tract Infection (UTI) | Have you experienced: | Either (I. and II.) OR, three or more of I-V. | SIGN[b] |
| | I. Pain passing urine | | |
| | II. Urinary frequency–passing urine more often | | |
| | III. Urinary urgency–need to pass urine quickly/difficulty in holding urine | | |
| | IV. Fever | | |
| | V. Abdominal pain | | |
| Perineal wound infection | At the site of a perineal wound (cut or tear in the vagina) have you experienced: | Either, I. OR, (IV AND one or both of II and III.) | CDC[a] |
| | I. Pus discharge | | |
| | II. Pain | | |
| | III. Swelling | | |
| | IV. Wound breakdown (wound edges separated) | | |
| Endometritis | Have you experienced: | Two or more of I-III where II is not explained by UTI and III is not explained by perineal wound infection. | CDC[a] |
| | I. Fever | | |
| | II. Abdominal pain | | |
| | III. Foul-smelling or pus vaginal discharge | In women with caesarean section this was counted as an organ space SSI | |
| Mastitis | Have you experienced: | Either, I. OR, both II. and III. | CDC[a] |
| | I. Swollen, hard area of the breast | | |
| | II. Painful, red breast | | |
| | III. Fever | | |
| pSBI | Has your baby experienced: | One or more of I-VII. | YICSS[c] |
| | I. Fever | | |
| | II. Very cold (low temperature) | | |
| | III. Very fast breathing | | |
| | IV. Chest indrawing (sucking in the ribs when breathing) | | |
| | V. Convulsions/fits | | |
| | VI. Poor feeding/not feeding | | |
| | VII. Only moving when stimulated | | |
| Umbilical cord infection | Has your baby experienced: | One or both of I. and II. | CDC[a] |
| | I. Redness around the umbilical cord stump | | |
| | II. Pus discharge from umbilical cord stump | | |

a)Centres for Disease Control [18]

b) Scottish Intercollegiate Guidelines Network [17]

c)Young Infants Clinical Signs Study [7].

or possible depression (6–30). Maternal function questions were analysed individually as "any" or "no difficulty" in performing the function.

Duplicate ID numbers and data entry errors were corrected where possible using hospital case-notes or comparing with other study data. Any remaining discordant data was dropped. There was inconsistency in the occurrence of stillbirths between data sources, therefore still-births were not analysed. Data on twin and triplet newborns was also inconsistent and in addition an error in ODK programming meant only data from the first baby was useable.

## Statistical analysis

Women's demographic and pregnancy data was described by delivery mode. Rates of infection were calculated from delivery until the day-28 telephone call using reported days from delivery to start of symptoms. Symptoms reported at both day-7 and day-28 were counted as distinct infection events if they started over 14 days apart, or if they met criteria for different infection types and started over seven days apart, or if initial symptoms had resolved by the day-7 inter-view. Date of death and infection data were not collected from babies who died before the day-7 interview, therefore these babies were excluded from infection outcome analyses. Babies who died after the day-7 interview contributed to infection analyses up to day 7. Using Cox regres-sion with robust standard errors to account for clustering by person, we explored associations between potential risk factors and the rate of maternal postnatal infection or possible newborn infection. Proportional hazards assumptions were checked using tests based on Schoenfeld Residuals. Factors showing evidence of association in the crude analysis (p<0.1) were explored further in multivariable models. Maternal age and delivery hospital were considered *a priori* confounders for risk of maternal postnatal infection. We restricted the parameters in the final models to 10% of the number of outcomes. For missing risk-factor data, we carried out multi-ple imputation using chained equations (MICE) because most variables were categorical, cre-ating 10 imputed datasets. Delivery mode and hospital were included as auxiliary variables. Women whose case-notes were missing were excluded from risk-factor analysis.

We report the highest level of care sought by women and newborns with possible infection and the percentage readmission to hospital for those with and without infection. We describe maternal depression and function at day-28 and explore associations with early postnatal infection using chi-squared tests and logistic regression.

## Ethics

The study was approved by the Tanzanian National Institute for Medical Research, IHI Insti-tutional Research Board and LSHTM Research Ethics Committee. Written informed consent was obtained from women on the postnatal wards. Willingness to continue in the study was confirmed at the start of each telephone interview. There was no public or patient involvement in the study design or interpretation of results. The Soapbox Collaborative supported the study following external peer review of the study proposal.

## Results

Between 15th March and 9th May 2018, research nurses recruited 879 women into the study, sam-pling from a total of 2,110 deliveries (18% caesarean section) (Fig 1). We interviewed 791 (90%) women at least once, providing data until day 7, and 753 (86%) completed the day-28 interview. Final interview occurred between 7 and 43 (median 29) days after delivery. Most women whose only interview was at day-28, reported that their telephone battery was not charged at day-7.

Case-notes were not located for 39 women. In the remaining 840, missing data was minimal except gestational age (39%). Mean age was 25 (range 18–45) years. Fewer than 3% of women

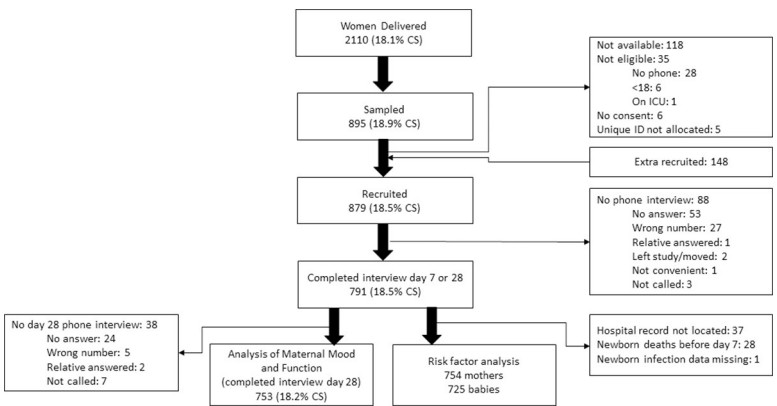

**Fig 1. Flow diagram.**

were referred-in. Induction and augmentation of labour, including artificial rupture of membranes, were uncommon (each <3%) but occurred more frequently at Amana Hospital than Temeke Hospital (S2 Table). Among vaginal births (n = 692), seven were breech and three were by vacuum extraction. Vaginal tears were experienced after 36% of vaginal deliveries and episiotomy was rare (Table 2). Among 829 liveborn babies, bag-and-mask resuscitation and admission were more common both following caesarean section and at Amana Hospital (Table 2, S1 Table). Average length of stay after delivery was 0.8 days following vaginal delivery (range 0–8) and 2.4 days post-caesarean section (range 0–7).

Antenatally, 7.4% of women received antibiotics, primarily for prophylaxis before caesarean section or following PROM. Postnatally, 62% of all women were prescribed antibiotics: 94% of women undergoing caesarean section and 98% of all women giving birth at Amana hospital were prescribed antibiotics (Table 3).

## Infection risk and rate

No postnatal maternal infections were documented in hospital case-notes at the time of birth and there were no maternal deaths. Among all 791 women with at least one telephone interview, 47 (5.9%) reported possible postnatal infection starting day 0–7. Symptoms of UTI affected 22 (2.8%) women and symptoms of endometritis affected 12 (1.5%). Among 146 women with caesarean section, 15 (10.3%) reported possible postnatal infection of whom 12 (8.2%) had symptoms of SSI (Table 4). From day 8–28, 9/753 (1.2%) developed possible postnatal infection. The rate of possible infection was 79.4 (95% Confidence Interval (CI) 61.1–103.2) per 1000 women per month.

Before the first interview, 28 (3.5%) babies were stillborn or died and one was missing infection data. Of the remaining 762 babies, 51 (6.7%) developed possible newborn infection from day 0–7, almost entirely attributable to pSBI (47, 6.2%) (Table 4). From day 8–28, another six babies died and 30/719 (4.3%) babies developed possible infection, one of whom had two episodes of infection. The rate of possible infection was 121.1 (95% CI 97.5–150.3) per 1000 babies per month. Three of these babies were diagnosed with sepsis in the maternal case-notes. For two of these three cases, no infection symptoms were reported by the mother at telephone interview.

Women sought care in a health facility following 37/56 (66%) episodes of possible postnatal infection: 24 (43%) at their delivery hospital, 8 (14%) at another hospital, and 5 (9%) at a lower level health facility. Babies were taken to a health facility following 72/82 (88%) episodes of

**Table 2. Demographic, pregnancy and newborn factors by mode of delivery for 840 women and 829 liveborn babies with maternal hospital case-notes.**

| | Vaginal Delivery n(%) (N = 692) | Caesarean Section n(%) (N = 148) | Total |
| --- | --- | --- | --- |
| | | | n(%) (N = 840) |
| Maternal age in years | | | |
| 18–24 | 288 (41.6) | 50 (33.8) | 338 (40.2) |
| 25–29 | 193 (27.9) | 42 (28.4) | 235 (28.0) |
| 30+ | 197 (28.5) | 52 (36.1) | 249 (29.6) |
| Missing | 14 (2.0) | 4 (2.7) | 18 (2.1) |
| Parity | | | |
| Nulliparous | 234 (33.8) | 52 (35.1) | 286 (34.1) |
| 1 | 205 (29.6) | 50 (33.8) | 255 (30.4) |
| 2 | 125 (18.1) | 23 (15.5) | 148 (17.6) |
| 3+ | 106 (15.3) | 19 (12.8) | 125 (14.9) |
| Missing | 22 (3.2) | 4 (2.7) | 26 (3.1) |
| Preterm birth (<37 weeks gestation) | 59 (8.5) | 22 (14.9) | 81 (9.6) |
| Missing | 287 (41.5) | 42 (28.4) | 329 (39.2) |
| Hypertensive disorders[a] | 18 (2.6) | 16 (10.8) | 34 (4.1) |
| Missing | 4 (0.6) | 2 (1.4) | 6 (0.7) |
| HIV | 29 (4.2) | 8 (5.4) | 37 (4.4) |
| Missing/not available | 14 (2.0) | 1 (0.7) | 15 (1.8) |
| PROM | 25 (3.6) | 4 (2.7) | 29 (3.5) |
| Missing | 2 (0.3) | 1 (0.7) | 3 (0.4) |
| Episiotomy | 10 (1.5) | NA | 10 (1.2) |
| Missing | 14 (2.0) | NA | 14 (1.7) |
| Perineal tear | 250 (36.1) | NA | 250 (29.8) |
| Missing | 3 (0.4) | NA | 3 (0.4) |
| PPH | 7 (1.0) | 2 (1.4) | 9 (1.1) |
| Missing | 2 (0.3) | 0 | 2 (0.2) |
| Antibiotics in labour | 26 (3.8) | 36 (24.3) | 62 (7.4) |
| Missing | 5 (0.7) | 1 (0.7) | 6 (0.7) |
| Antibiotics postpartum | 382 (55.2) | 139 (93.9) | 521 (62.0) |
| Missing | 5 (0.7) | 3 (2.0) | 8 (1.0) |
| **Newborn Factors** | **Vaginal (N = 681)** | **CS (N = 148)** | **Total (N = 829)** |
| Apgar Score at 5 minutes <7 | 5 (0.7) | 5 (3.4) | 10 (1.2) |
| Missing | 2 (0.3) | 1. (0.7) | 3 (0.4) |
| Bag and mask resuscitation | 9 (1.3) | 8 (5.4) | 17 (2.1) |
| Missing | 2 (0.3) | 2 (1.4) | 4 (0.5) |
| Admission | 10 (1.5) | 12 (8.1) | 22 (2.7) |
| Missing | 0 | 2 (1.4) | 2 (0.2) |

[a]Hypertensive disorders: 2 eclampsia, 19 pre-eclampsia, 17 pregnancy-induced hypertension.

possible infection: 38 (46%) to the delivery hospital, 25 (30%) to another hospital, and 9 (11%) to a lower level health facility.

## Associations with infection

There was evidence that caesarean delivery doubled the rate of possible maternal postnatal infection compared to women who had a vaginal delivery, and this association remained after adjusting for maternal age and hospital (adjusted Hazard Ratio (aHR) 1.93, 95% CI 1.11–3.36,

**Table 3. Reason for antibiotics prescribed to women in hospital during labour and postpartum by delivery mode.**

| | Vaginal Delivery n (%) | Caesarean Section n (%) | Total |
|---|---|---|---|
| **Antibiotics in labour** | **N = 26** | **N = 36** | **N = 62** |
| Caesarean section prophylaxis | 0 | 34 (94.4) | 34 (54.8) |
| PROM | 14 (53.9) | 2 (5.6) | 16 (25.8) |
| UTI | 1 (3.9) | 0 | 1 (1.6) |
| Other | 8 (30.8) | 0 | 8 (12.9) |
| None | 3 (11.5) | 0 | 3 (4.8) |
| **Antibiotics postpartum** | **N = 382** | **N = 139** | **N = 521** |
| Caesarean section prophylaxis | 0 | 131 (94.2) | 131 (25.1) |
| PROM | 2 (0.5) | 0 | 2 (0.4) |
| Perineal suture | 172 (45.0) | 0 | 172 (33.1) |
| UTI | 1 (0.3) | 0 | 1 (0.2) |
| Routine | 190 (49.7) | 0 | 190 (36.5) |
| IUD | 4 (1.1) | 0 | 4 (0.8) |
| Unknown/not recorded | 13 (3.4) | 8 (5.8) | 21 (4.0) |

p = 0.02). There was also weak evidence of an association between women's age-group and infection (p = 0.06) with the highest infection rates occurring in women aged 25–29 (Table 5).

Bag-and-mask resuscitation at birth was strongly associated with possible newborn infection compared to babies who were not resuscitated (aHR 4.45, 95% CI 2.10–9.44, p<0.001), however this was a rare exposure (n = 11 babies). There was weak evidence for increased possible newborn infection if the mother received antibiotics in labour compared to mothers who did not (Table 6).

In the first seven days postnatal 7/762 mother-baby pairs both experienced possible infection. Mother's with postnatal infection in the first 7 days had an increased risk of their baby suffering possible newborn infection during this time period, compared to mother's without infection (crude Odds Ratio 2.74, 95%CI 1.16–6.48, p = 0.02).

## Consequences of infection

At the day-7 interview, 5/43 (12%) women with possible postnatal infection reported they had been readmitted to hospital as compared with only 5/696 (0.7%) women without infection. All women readmitted with infection had given birth by caesarean section. Among 713 babies

**Table 4. Maternal and newborn infections occurring up to 7 days after delivery.**

| Maternal infection | Vaginal delivery n(%) N = 645 | Caesarean section n(%) N = 146 | Total n(%) N = 791 |
|---|---|---|---|
| Postnatal infection | 32 (5.0) | 15 (10.3) | 47 (5.9) |
| Endometritis | 12 (1.9) | NA | 12 (1.5) |
| SSI | NA | 12 (8.2) | 12 (1.5) |
| Perineal wound infection | 7 (1.1) | 0 | 7 (0.9) |
| UTI | 15 (2.3) | 7 (4.8) | 22 (2.8) |
| Mastitis | 13 (2.0) | 3 (2.1) | 16 (2.0) |
| **Newborn infection** | **N = 621** | **N = 141** | **N = 762** |
| Possible newborn infection | 40 (6.4) | 11 (7.8) | 51 (6.7) |
| pSBI | 36 (5.8) | 11 (7.8) | 47 (6.2) |
| Umbilical cord infection | 5 (0.8) | 0 | 5 (0.7) |

**Table 5. Association between potential risk-factors and rate of possible maternal postnatal infection.**

| Factor | Total women | Episodes of postnatal infection | Person-time (months) | Rate of infection per 1000 person months | Crude HR (95% CI) N = 754[a] | Wald p-value | Adjusted HR (95% CI) N = 754 [a] | Wald p-value |
|---|---|---|---|---|---|---|---|---|
| All women | 791 | 56 | 705.3 | 79.4 (61.1–103.2) | | | | |
| Delivery mode | | | | | | | | |
| Vaginal | 645 | 39 | 578.1 | 67.5 | 1 | 0.02 | 1 | 0.02 |
| Caesarean section | 146 | 17 | 127.3 | 133.6 | 1.95 (1.12–3.37) | | 1.93 (1.11–3.36) | |
| Maternal age (years) | | | | | | | | |
| 18–24 | 303 | 15 | 167.9 | 56.0 | 1 | 0.05 | 1 | 0.06 |
| 25–29 | 212 | 23 | 186.0 | 123.6 | 2.20 (1.15–4.28) | | 2.14 (1.12–4.09) | |
| 30+ | 223 | 16 | 204.3 | 78.3 | 1.43 (0.72–2.84) | | 1.37 (0.69–2.70) | |
| Hospital | | | | | | | | |
| Amana | 403 | 28 | 362.0 | 77.4 | 1 | 0.87 | 1 | 0.98 |
| Temeke | 388 | 28 | 343.4 | 81.5 | 1.04 (0.62–1.75) | | 1.01 (0.60–1.70) | |
| Parity | | | | | | | | |
| 0 | 252 | 15 | 224.8 | 66.7 | 1 | 0.81 | | |
| 1 | 233 | 19 | 206.9 | 91.8 | 1.33 (0.69–2.56) | | | |
| 2 | 131 | 11 | 115.0 | 95.7 | 1.37 (0.65–2.89) | | | |
| 3+ | 113 | 8 | 103.6 | 77.2 | 1.16 (0.48–2.81) | | | |
| Preterm birth (<37 weeks) | | | | | | | | |
| No | 392 | 30 | 346.5 | 86.6 | 1 | 0.89 | | |
| Yes | 69 | 5 | 62.5 | 80.1 | 0.94 (0.37–2.35) | | | |
| Antibiotics in labour | | | | | | | | |
| No | 697 | 48 | 622.6 | 77.1 | 1 | 0.17 | | |
| Yes | 51 | 6 | 43.7 | 137.3 | 1.75 (0.78–3.91) | | | |
| Postpartum antibiotics | | | | | | | | |
| No | 277 | 18 | 246.4 | 73.0 | 1 | 0.49 | | |
| Yes | 469 | 37 | 417.9 | 88.5 | 1.22 (0.69–2.16) | | | |

[a]Values imputed for variables with missing data, except for Preterm birth where the amount of missing data was considered too large to impute.

Results shown if >2 infections in a single category. Full results in S2 Table.

alive at the day-7 interview, 44% with possible infection had been readmitted to hospital compared with 1.8% of those without.

Depression scores ranged from 0–10/30 among 753 women at day-28 interview and 31 (4%) had possible depression (score > = 6). Among 43 women with early postnatal infection (day 0–7), 4 (9.3%) developed possible depression versus 27 (3.8%) of those without infection (OR 2.1, 95% CI 0.64–6.89, p = 0.22, adjusting for death of the baby).

**Table 6. Association between potential risk factors and rate of possible newborn infection.**

| Factor | Total newborns | Episodes of possible infection | Person-time (months) | Rate of infection per 1000 person months | Crude HR (95% CI) N = 725 [a] | Wald p-value | Adjusted HR (95% CI) N = 725 [a] | Wald p-value |
|---|---|---|---|---|---|---|---|---|
| All babies | 762 | 82 | 677.4 | 121.1 (97.5–150.3) | | | | |
| Resuscitation (bag and mask) | | | | | | | | |
| No | 709 | 75 | 629.9 | 119.1 | 1 | <0.001 | 1 | <0.001 |
| Yes | 11 | 5 | 8.7 | 574.3 | 4.61 (2.35–9.04) | | 4.45 (2.10–9.44) | |
| Antibiotics in labour | | | | | | | | |
| No | 674 | 69 | 598.5 | 115.3 | 1 | 0.01 | 1 | 0.08 |
| Yes | 47 | 10 | 39.9 | 250.9 | 2.15 (1.18–3.91) | | 2.00 (0.93–4.30) | |
| Delivery mode | | | | | | | | |
| Vaginal | 621 | 64 | 552.2 | 115.9 | 1 | 0.35 | 1 | 0.95 |
| Caesarean section | 141 | 18 | 125.1 | 143.8 | 1.24 (0.74–2.09) | | 1.02 (0.55–1.91) | |
| PROM | | | | | | | | |
| No | 698 | 75 | 617.7 | 121.4 | 1 | 0.37 | 1 | 0.76 |
| Yes | 24 | 4 | 22.1 | 180.6 | 1.53 (0.61–3.84) | | 1.16 (0.45–2.99) | |
| Maternal age (years) | | | | | | | | |
| 18–24 | 291 | 29 | 256.9 | 112.9 | 1 | 0.51 | | |
| 25–29 | 203 | 27 | 280.4 | 149.6 | 1.34 (0.79–2.28) | | | |
| 30+ | 216 | 22 | 193.0 | 114.0 | 1.05 (0.60–1.84) | | | |
| Hospital | | | | | | | | |
| Amana | 388 | 41 | 347.1 | 118.1 | 1 | 0.94 | | |
| Temeke | 374 | 41 | 330.3 | 124.1 | 1.04 (0.67–1.61) | | | |
| Preterm (<37 weeks gestation) | | | | | | | | |
| No | 376 | 38 | 330.6 | 114.9 | 1 | 0.65 | | |
| Yes | 67 | 8 | 59.5 | 134.5 | 1.18 (0.57–2.44) | | | |
| Postpartum antibiotics | | | | | | | | |
| No | 266 | 21 | 236.8 | 88.7 | 1 | 0.07 | | |
| Yes | 452 | 58 | 399.5 | 145.2 | 1.59 (0.96–2.62) | | | |

[a]Values imputed for variables with missing data, except for Preterm birth where the amount of missing data was considered too large to impute.

Results not shown if <3 infections in a single category. Full results in S3 Table.

At day-28 interview, 103/752 (13.7%) women reported difficulty with housework and 8/751 (1.1%) reported difficulty washing themselves. Among women with a living baby, 43/718 (6.0%) reported difficulty carrying or caring for their baby and 99.7% were exclusively breast-feeding. Difficulty with each activity was reported more frequently among women with possible early postnatal infection compared to those without infection, but statistical evidence was inconsistent (Table 7).

**Table 7. Associations between early maternal postnatal infection (day 0–7) and maternal function at day 28.**

|  | Difficulty washing | Difficulty with housework | Difficulty carrying baby | Difficulty caring for baby |
|---|---|---|---|---|
|  | n/N (%) | n/N (%) | n/N (%) | n/N (%) |
| Postnatal infection |  |  |  |  |
| No | 6/709 (0.9) | 94/709 (13.3) | 39/679 (5.7) | 38/679 (5.6) |
| Yes | 2/42 (4.8) | 9/43 (20.9) | 4/39 (10.3) | 5/39 (12.8) |
| Chi$^2$ p-value | 0.02 | 0.16 | 0.25 | 0.06 |

## Discussion

We conducted telephone interviews with 791 women at seven and/or 28 days after hospital childbirth in Dar es Salaam, Tanzania. We estimated a rate of 79.4 possible maternal and 121.1 possible newborn infections per 1000 person-months. Women with caesarean birth had twice the rate of infection. Newborns resuscitated at birth had over four times the rate of infection. Women and newborns with possible infection had substantially higher readmission rates compared with those without infection, and there was a trend towards increased depression risk following early infection. Telephone surveillance proved feasible: 97% of the initial sample had access to a mobile telephone and 90% of all recruited women were interviewed at least once.

Global incidence of pregnancy-related infection estimated by the Global Burden of Disease study 2017 equates to 8.2% of livebirths [19], and the recent Global Maternal Sepsis Study (GLOSS) reports prevalence of infection in hospitalised pregnant and postpartum women of 70.4 per 1000 livebirths [10]; however, their broader case definitions prevent direct comparison with our study. Our incidence of endometritis at day-7 (1.5%) is consistent with the 1.7% (95% CI 1.4–2.1%) estimate for SSA from a recent meta-analysis [6]. However, we observed a caesarean surgical site infection risk of 8.2%, which is lower than the 15.6% estimate from a systematic review for SSA [20]. Our incidence of pSBI (6.2%) was the same as the estimate for SSA from a meta-analysis of studies in which health or community workers applied YICSS criteria [21].

Caesarean section is an established risk factor for maternal infection and sepsis [9, 10, 22] and in our study carried a higher risk of both SSI and UTI than vaginal birth. Increasing rates of caesarean childbirth and evidence of antimicrobial resistance (AMR) in subsequent infections [23] demand enhanced infection prevention measures. Pre-operative prophylactic antibiotics are effective [24] and recommended in Tanzania [25], but were documented before only 24% of caesarean sections. Newborn infection could result from pathogens introduced during resuscitation, explaining the strong association seen. Additionally, sick newborns requiring ventilation are at increased risk of infection, supporting calls to improve both intrapartum care and postnatal infection prevention [26].

Expected associations between prematurity, PROM, PPH, HIV, and either maternal or newborn infection were not evident, but these factors were reported less frequently than expected. Induction and augmentation of labour were similarly infrequent. This could reflect poor documentation at the hospitals or difficulties in extraction. Postpartum antibiotics were not associated with reduced infection incidence, providing no justification for universal prescribing observed at one study hospital. This practice is not recommended nationally or internationally [27], could be a driver of AMR and needs to be challenged. There was some evidence of a crude association between maternal and newborn infection, also found in a systematic review of maternal infection in labour [28], suggesting a shared aetiology for some infections and highlighting the importance of caring for the woman and newborn synergistically.

Depression prevalence (4.1%) was lower than other LMIC studies that also used EPDS at 4–8 weeks postnatal. However, these studies showed considerable heterogeneity

(range 4.9–50.8%) [29]. Telephone follow-up could provide a valuable tool to screen for post-natal depression and warrants further validation. We did not power our study to assess associations between maternal infection and depression or functioning, but our results suggest a trend in that direction, compatible with previous studies of maternal morbidity [29–32].

In our study, 66% of women and 88% of newborns with possible infection had sought health-facility care when interviewed, revealing the important proportion of cases that would be missed by a purely hospital based study. Telephone diagnosis of caesarean site infection achieved high specificity in Kenya and Tanzania [12, 13]. Telephone surveillance detected more cases of SSI than using patient case-notes or written surveys in high-income settings [33, 34]. Mobile telephone access was high in our study sample (97%), and we reached a high proportion of recruited women (90%), supporting the feasibility of telephone surveillance in comparable LMIC settings.

## Strengths and limitations

Our study benefited from collecting data on specific components of standard infection definitions during the interview that were used in diagnosis algorithms, rather than relying on women's or data collectors' judgement. We collected data with a short recall period, reducing potential bias, and used symptom start dates to show infection distribution over time and estimate incidence rate. Although we recruited from two tertiary hospitals, we expect the population to be broadly representative of Dar es Salaam region where 94% of women are estimated to give birth in a facility and 17% by caesarean, similar to our study population.

The main limitation of this study is the unknown validity of the questionnaire to identify true cases of infection. We believe that the substantially increased rates of hospital readmission amongst women and newborns with telephone-based diagnosis of infections provide strong post-hoc support for the validity of our approach. Incidence of endometritis and pSBI and the association with caesarean childbirth are all closely consistent with other studies, lending further support to the results. However, we identified fewer cases of SSI than other studies, and we had two cases of neonatal sepsis extracted from hospital case-notes that were not subsequently reported at maternal interview. In addition, newborn deaths from infection were not captured, therefore true infection incidence may be higher than estimated. Furthermore, hospital case-notes were not located for 39 women, in some cases following admission of the baby, potentially reducing estimated infection incidence. It is possible that women who were unwell, or caring for a sick baby, were less likely to answer their telephones, also leading to an under-estimate of infection incidence. However, the use of a second telephone number belonging to a friend/relative, the repeated call attempts over seven days and the second interview at day-28 reduce this risk.

## Conclusion

Our telephone surveillance study found a substantial and plausible rate of possible infection among mothers and newborns in urban Tanzania in the first month postnatal. Telephone interviews were feasible and identified cases that could be missed by hospital data collection alone. Results were consistent with previous studies, although further validation studies are needed. Therefore, this method of data collection shows promise for further use, both as a research tool and for routine medical practice. This could be of particular benefit during the current COVID pandemic, with concerns about reduced hospital attendance and the encouragement to work remotely. WHO does not recommend the use of routine postpartum antibiotics. Their use in this context showed no benefit and should be challenged. However, better implementation of pre-operative antibiotic prophylaxis for caesarean section is urgently required to mitigate the infection risk in mothers.

## Supporting information

**S1 Appendix. Telephone questionnaire–Day 28.**
(DOCX)

**S2 Appendix. Hospital case-note extraction form.**
(DOCX)

**S1 Table. Demographic, pregnancy and newborn factors by mode of delivery for 840 women and 829 liveborn babies with hospital record data by study hospital.**
(DOCX)

**S2 Table. Associations between potential risk factors and possible maternal postnatal infection.**
(DOCX)

**S3 Table. Associations between potential risk factors and possible newborn infection.**
(DOCX)

## Acknowledgments

With thanks to the four research nurses who recruited women to the study and extracted their hospital data; all members of the CLEAN study team at IHI and LSHTM for providing logistical support; the hospital management and medical and nursing staff in the maternity units at Amana and Temeke hospitals for agreeing to the study, providing space to recruit women and giving access to hospital case-notes; and most of all the women who participated.

## Author Contributions

**Conceptualization:** Susannah L. Woodd, Oona M. R. Campbell, Alexander M. Aiken, Wendy J. Graham.

**Formal analysis:** Susannah L. Woodd, Andrea M. Rehman.

**Funding acquisition:** Susannah L. Woodd, Wendy J. Graham.

**Investigation:** Abdunoor M. Kabanywanyi, Asila Kagambo, Warda Martiasi.

**Methodology:** Susannah L. Woodd, Oona M. R. Campbell, Louise M. TinaDay, Alexander M. Aiken, Wendy J. Graham.

**Project administration:** Susannah L. Woodd, Abdunoor M. Kabanywanyi.

**Supervision:** Susannah L. Woodd.

**Writing – original draft:** Susannah L. Woodd.

**Writing – review & editing:** Susannah L. Woodd, Abdunoor M. Kabanywanyi, Andrea M. Rehman, Oona M. R. Campbell, Asila Kagambo, Warda Martiasi, Louise M. TinaDay, Alexander M. Aiken, Wendy J. Graham.

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
