## [Decision Letter · Decision Letter 0]

15 Apr 2021

PONE-D-21-01978

Postnatal infection surveillance by telephone in Dar es Salaam, Tanzania: an observational cohort study

PLOS ONE

Dear Dr. Woodd,

Thank you for submitting your manuscript to PLOS ONE. After careful consideration, we feel that it has merit but does not fully meet PLOS ONE’s publication criteria as it currently stands. Therefore, we invite you to submit a revised version of the manuscript that addresses the points raised during the review process.

We look forward to receiving your revised manuscript.

Kind regards,

Calistus Wilunda, DrPH

Academic Editor

PLOS ONE

Additional Editor Comments:

Footnote ‘b’ in tables 5 and 6 is not clear. Moreover, I could not identify footnote 'b' in Table 6. 

Please use a consistent style for footnotes. In table 2, a special character is used but elsewhere letters are used.

RR can mean Risk ratio or rate ratio. Please specify. Given that you used Cox regression, why not report the effects as hazard ratios.

In reporting the results of association, especially in the abstract, please state explicitly the comparison group.

Journal Requirements:

Reviewers' comments:

Reviewer's Responses to Questions

**Comments to the Author**

1. Is the manuscript technically sound, and do the data support the conclusions?

Reviewer #1: Yes

Reviewer #2: Yes

2. Has the statistical analysis been performed appropriately and rigorously? 

Reviewer #1: Yes

Reviewer #2: Yes

3. Have the authors made all data underlying the findings in their manuscript fully available?

Reviewer #1: Yes

Reviewer #2: Yes

4. Is the manuscript presented in an intelligible fashion and written in standard English?

Reviewer #1: Yes

Reviewer #2: Yes

5. Review Comments to the Author

Reviewer #1: This study estimates rates of postnatal infection among women and infants in Dar es Salaam, Tanzania, using telephone surveillance. It demonstrates the feasibility of telephone surveillance for this purpose in Tanzania, and provides the first post-discharge estimates of maternal and newborn infection rates in this setting, as well as associated risk factors. It is clear, well-written, the methods and conclusions are sound, and it is a valuable addition to the literature on maternal and newborn health in sub-Saharan Africa.

Minor comments:

Line 72: Can any more information about the selected hospitals be provided? Are they public/private? What is their catchment area? How many hospitals of this size are there in Dar es Salaam?

Line 92: Given that one of the major limitations of the study is around the ability of the questionnaire to accurately identify infections, could you provide more details here on how the standard definitions were adapted? Are there any existing questionnaires for measuring postnatal infection that were considered but not used? If so, it would be helpful to discuss this here too.

Line 128: What was the frequency of discordant data, stillbirths and twins/triplets?

Reviewer #2: Line 203 - 28 early stillbirths occurred, did you include these in your final analysis?

Line 204 - between day 8-28 additional 6 babies died, did you include these numbers in you final analysis?

Line 217 - What do you mean by "weak evidence" of higher infection rates in women 25-29 years ( you make reference to table 5: [2.14 (1.12- 4.09): p=0.06]. How do you explain these p-value versus CI above 1.0?

Line 235: Maternal postnatal infection was associated with increased odds of possible newborn infection (crude Odds Ratio 2.74, 95%CI 1.16–6.48, p=0.02). Author should rephrase this statement or provide some explanation as well as discuss this in the section for discussion of results

Line 238-239: Please rephrase to provide easy readability

6. PLOS authors have the option to publish the peer review history of their article (what does this mean?). If published, this will include your full peer review and any attached files.

Reviewer #1: No

Reviewer #2: **Yes: **Richard Mangwi Ayiasi

---

## [Author Response · Author response to Decision Letter 0]

8 Jun 2021

Response to Editor's comments

The competing interests statement is updated as follows: 

"I have read the journal's policy and the authors of this manuscript have the following

competing interests: SLW, AMK, AK, WM and WJG received personal salary funded by

The Soapbox Collaborative for their work on the study. SLW received a Research

Degree Travel Grant from LSHTM. AMR received salary from the UK Medical Research Council and the UK Department for International Development (DFID) under the MRC/DFID Concordat agreement. Other authors declare no competing interests. This does not alter our adherence to PLOS ONE policies on sharing data and materials."

In tables 5 and 6 we have now combined the information into a single footnote:

Values imputed for variables with missing data, except for Preterm birth where the amount of missing data was considered too large to impute

We hope it is now clear that imputation was carried out for all variables except Preterm birth

We have updated table 2 to use a letter instead of a symbol for the footnote.

As suggested, we have changed the reporting to hazard ratios

We have now explicitly stated the comparison group for each association, in both the abstract, and the results section

Response to reviewers' comments

Reviewer 1:

The hospitals are two of the three public regional referral hospitals in Dar es Salaam, covering the districts of Ilala and Temeke. This has been reflected in the text. 

For information, there are 28 hospitals at this level in Tanzania. 

We adapted the infection definitions by removing all laboratory investigations and any clinical signs that could not be easily reported by the women, for example, temperature readings, abdominal tenderness, perineal wound redness. We were less rigid with some definitions e.g. superficial wound infection, to allow for these necessary omissions. A more detailed account is being written up elsewhere. A short sentence has been added to the manuscript.

"These definitions were adapted by the first author to include only symptoms and signs easily reported by the women."

We did not search extensively for other questionnaires, but from our work in this field, including a systematic review of maternal peripartum infection, we are not aware of any published, (validated) telephone questionnaires for this purpose. 

Regarding twins and stillbirths: Among the 12 interviews with mothers of liveborn twins, only six record data on the second twin, partially due to a fault in the ODK programming.

Among mothers with hospital and telephone data, 28 have a baby who dies in the first 7 days. Of these, maternal cause-of-death reports are consistent with 7/8 of the hospital-recorded stillbirths. However, the hospital records miss 4 more stillbirths reported by mothers and at Temeke hospital no stillbirths are recorded, which we consider highly unlikely. 

Reviewer 2:

Women giving birth in hospital and recorded in the delivery register were eligible for recruitment, including those with stillbirths. All recruited women who were reached for interview were included in the analysis of maternal postnatal infection. 

Stillbirths contribute to the 28 deaths occurring up to day 7. Stillbirths were not included in the analysis of newborn infection.

As described in the methods, line 104, babies who died after day 7 contributed to the analysis (infection rate and analysis of associations) up to day 7 only.

Line 217 p=0.06: The p-value refers to the overall association between the three categories of age-group and infection rate rather than the 25-29 year group alone. We have rephrased the text to better reflect the result.

"There was also weak evidence of an association between women’s age-group and infection (p=0.06) with the highest infection rates occurring in women aged 25–29."

---

## [Editor Report · Decision Letter 1]

21 Jun 2021

Postnatal infection surveillance by telephone in Dar es Salaam, Tanzania: an observational cohort study

PONE-D-21-01978R1

Dear Dr. Woodd,

We’re pleased to inform you that your manuscript has been judged scientifically suitable for publication and will be formally accepted for publication once it meets all outstanding technical requirements.

Kind regards,

Calistus Wilunda, DrPH

Academic Editor

PLOS ONE
---

## [Editor Report · Acceptance letter]

23 Jun 2021

PONE-D-21-01978R1 

Postnatal infection surveillance by telephone in Dar es Salaam, Tanzania: an observational cohort study 

Dear Dr. Woodd:

I'm pleased to inform you that your manuscript has been deemed suitable for publication in PLOS ONE. Congratulations! Your manuscript is now with our production department. 

Kind regards, 

on behalf of

Dr. Calistus Wilunda 

Academic Editor

PLOS ONE